# The Relationship between Attitudes and Satisfaction Concerning the COVID-19 Vaccine and Vaccine Boosters in Urban Bangkok, Thailand: A Cross-Sectional Study

**DOI:** 10.3390/ijerph19095086

**Published:** 2022-04-22

**Authors:** Jadsada Kunno, Busaba Supawattanabodee, Chavanant Sumanasrethakul, Chuthamat Kaewchandee, Wachiraporn Wanichnopparat, Krit Prasittichok

**Affiliations:** 1Department of Research and Medical Innovation, Faculty of Medicine Vajira Hospital, Navamindradhiraj University, Bangkok 10300, Thailand; sbusaba@nmu.ac.th (B.S.); chuthamat@nmu.ac.th (C.K.); wachiraporn.w@nmu.ac.th (W.W.); krit.p@nmu.ac.th (K.P.); 2Department of Urban Medicine, Faculty of Medicine Vajira Hospital, Navamindradhiraj University, Bangkok 10300, Thailand; chavanant@nmu.ac.th

**Keywords:** COVID-19, vaccine booster, attitudes, satisfaction, Bangkok

## Abstract

Background: COVID-19 vaccine hesitancy is a global concern. Many individuals are concerned about the potential side-effects of the COVID-19 vaccine and vaccine boosters. The purpose of this study was to assess attitudes and satisfaction concerning COVID-19 vaccines and vaccine boosters in the population in Bangkok, Thailand. Methods: A cross-sectional online survey measuring COVID-19 vaccine attitudes and satisfaction was distributed from September to December 2021. Multiple linear regression was used to explore associations between demographic variables and questionnaire results. Spearman’s correlation analysis was used to examine associations between attitude and satisfaction scores. Results: A total of 780 questionnaire responses were obtained. The largest groups of participants reported having obtained a first vaccination dose via viral vaccine (52.8%), a second vaccination booster via viral vaccine (49.5%), and a third vaccination booster via mRNA vaccine (28.8%). Multiple linear regression revealed a lower association between vaccine attitude scores and having earned less than a bachelor’s degree (β −0.109; 95% CI −2.541, −0.451) and infection risk without self-isolating (β −0.154; 95% CI −4.152, −0.670) compared with attaining a bachelor’s degree or higher and never having being at risk of infection, respectively. Higher vaccine satisfaction scores were more closely associated with being married than being single (β 0.074; 95% CI −0.073, 3.022), whereas lower vaccine satisfaction scores were less closely associated with non-healthcare workers (β −0.143; 95% CI −4.698, −0.831) and infection risk without self-isolating (β −0.132; 95% CI −6.034, −0.502) compared with non-healthcare workers and never being at risk of infection. There was weak but significant positive correlation between attitude and satisfaction scores (*r* = 0.338, *p*-value < 0.001). Hence, a gradual decline in protection following vaccination and the positive effects of a booster dose after primary vaccination have made the decision to administer booster doses. Conclusion: The results suggest that policymakers need to develop more effective strategies to raise awareness about the importance of vaccination.

## 1. Introduction

Many countries have experienced multiple waves of coronavirus outbreaks. Most people infected with the COVID-19 virus will experience a mild to moderate respiratory illness and recover without requiring special treatment [1]. However, empirical data show that characteristics varied between waves during the 2020 pandemic [2]. As a result of anti-vaccine campaigns, distrust of vaccines has been increasingly reported during the last few years [3]. Many people are concerned about the potential side-effects of vaccines, and COVID-19 vaccination hesitancy has become a global concern. Previous studies indicate that the most common reasons cited for refusing the vaccine were insufficient knowledge about short- and/or long-term adverse effects, distrust of drug companies, and the belief that the virus is not dangerous [4]. Doubts and hesitations about vaccination can result in individuals delaying or rejecting vaccination [5]. There is a strong call to understand the roots of vaccine hesitancy and the drivers of vaccine acceptance in order to successfully combat the current pandemic. However, epidemiologists and public health educators face substantial challenges to their efforts to clarify this foggy scene and provide evidence-informed recommendations regarding COVID-19 vaccines and vaccine boosters [6].

Following reports of a gradual decline in protection following vaccination and the positive effects of a booster dose six months after primary vaccination, many countries have made the decision to administer booster doses [7,8,9]. However, individuals who have already been vaccinated may accept or reject a vaccine booster dose due to various reasons, including side effects experienced following the previous (primer) doses, the perceived effectiveness of the vaccine booster dose, the vaccinee’s susceptibility to the target infection, and other safety concerns [10,11].

Vaccine attitudes are influenced by a complex interaction of social, cultural, political, and individual factors. However, few studies have examined how these factors influence attitudes and satisfaction concerning COVID-19 vaccine or booster doses. To address this gap, this study explored associations between socio-demographic variables and COVID-19 vaccine booster attitudes and satisfactions as well as correlations between attitudes and satisfaction scores in a sample participant in Bangkok, Thailand.

## 2. Methods

### 2.1. Study Design

This study entailed the analysis of a cross-sectional survey distributed among participants in urban Bangkok, Thailand, from September to December 2021. The study was approved by the ethics committee of the Faculty of Medicine Vajira Hospital, Navamindradhiraj University, Bangkok, Thailand, which is in full compliance with the international guidelines for human research protection as set out in the Declaration of Helsinki, the Belmont Report, CIOMS Guidelines, and the International Conference on Harmonization in Good Clinical Practice (ICH-GCP) (COA 151/2564). The participants were informed that they could leave the study at any time without justifying their decision, and no data were saved before the participants finalized the questionnaire and confirmed submitting their answers. To maximize anonymity, no identifying personal data were collected from the participants.

### 2.2. Participants

Participants aged 18 years and older living in Bangkok were eligible to participate in the study. The sample size was calculated using G*Power based on the estimated population in the city. The sample has been more than expected, based on online survey. Thus, the sample size of 780 participants was proposed to provide support for external validity on the results of the study.

### 2.3. Data Collection

Data were collected using an online survey (Google form, including consent form) distributed on social media using a snowball technique. The invitation asked participants to confirm that informed consent was obtained from all subjects, voluntary participation and provided instructions for filling in the questionnaire. All participants have been performed in accordance with the Declaration of Helsinki and have been approved by an appropriate ethics committee.

### 2.4. Questionnaire

The questionnaire takes about 10 min to complete and is divided into 4 sections. The questions were designed and modified by an expert team of researchers. The first section collected socio-demographic information, including age, gender, marital status, education, occupation, exposure to COVID-19, and vaccination history (see Table 1 for details). Content validity was examined by three experts. Cronbach’s alpha coefficient for attitudes showed 0.91 and content validity 0.87. For satisfaction, the item content validity score was 0.93 and the Cronbach’s alpha coefficient was 0.751.

The attitudes section consists of 17 items (A1–A14) assessing perceptions of whether the country could win the fight against the COVID-19 pandemic. The attitude section consists of 14 items scored as Agree = 3, not sure = 2, and disagree = 1. Attitude item responses were summed to a total score ranging from 0–42 and classified into 3 categories, namely, favorable attitudes (33–42), medium favorable attitudes (24–32), and unfavorable attitudes (14–23) (see Table 2 for details). The satisfaction section consists of 15 items scored as very satisfied = 3, medium satisfied = 2, and low satisfied = 1. Satisfaction item responses were summed to a total score ranging from 0–45 and classified into 3 categories, namely, low satisfaction (15–25), medium satisfaction (26–35), and high satisfaction (36–45; see Table 2 for details).

### 2.5. Statistical Analysis

Descriptive statistics including frequency, percentages, means, and standard deviations were analyzed to describe participants’ socio-demographic characteristics and attitude and satisfaction item scores. Multiple linear regression was used to explore associations between socio-demographic variables and attitude and satisfaction scores. Spearman’s correlation analysis was used to examine associations between attitude and satisfaction scores. The level of statistical significance was set at *p*-value < 0.05. The statistical analysis was performed using the Statistical Package for the Social Sciences Program (SPSS), version 22 (IBM Corp., Armonk, NY, USA).

## 3. Results

A total of 780 questionnaire responses were obtained, and socio-demographic and vaccination history are presented in Table 1.

The largest group of participants (75.9%) was female, aged ≤ 42 years old (50.6%), single (51.2%), had attained a bachelor’s degree (61.1%), worked as HCWs (49.6%), and had no diseases (64.1%). The majority had never been exposed to COVID-19 (43.5%). Among vaccine types, the largest groups had obtained their first and second vaccine doses via viral vaccine (52.8% and 49.5%, respectively) and a third booster dose via mRNA vaccine (28.8%). However, 2.7% and 53.1% of participants reported that they had not received a second or third vaccine booster dose, respectively. These results indicate that the population was representative of the wider population of population residing in Bangkok.

### 3.1. Attitude and Satisfaction Scores

Table 2 summarizes participants’ attitude and satisfaction item scores. The mean attitudes score was (34.63 ± 6.14). The majority (90.5%) of participants agreed with the state’s recommendation to receive two full doses of the vaccine and that the best prevention practice is to always use a mask and clean the hands with soap and water or alcohol gel and practice social distancing (A2). Most participants agreed that obtaining accurate information about the COVID-19 vaccine would influence their decision on the type of vaccine obtained (A6), that members of the general public in at-risk areas should receiving a third vaccine dose (A11), and that all Thai people should receive the third vaccine dose due to COVID-19 mutations (A13). Approximately 41% were not sure that government measures would be effective in controlling the spread of COVID-19 (A10), and 48.8% were not sure that people living in at-risk areas should be vaccinated with a viral vector vaccine for the third dose (A14).

The mean satisfaction score was (26.37 ± 9.69). The majority of participants expressed high satisfaction with selecting the viral vector vaccine (S5) and with the type and efficacy of the viral vector vaccine (S13). Most expressed medium satisfaction with the quality of the allocated vaccines (40.8%) (S3), that the registration for injection of viral vector vaccines would be appointed on time (S9), and with the type and efficacy of the inactivated vaccine (S12). The largest groups expressed low satisfaction with inactivated vaccines (S1) and that the inactivated vaccine was sufficient for the Thai population (S6). Similarly, the largest groups reported low satisfaction with the viral vector vaccine and that the viral vaccine was sufficient for the Thai population (S7). Most participants indicated low satisfaction that vaccine allocations were being distributed across all areas (S10) as well as with the role of government credibility in procuring and administering vaccines (S11).

### 3.2. Associations between Socio-Demographic Variables and Attitude and Satisfaction Scores

As Table 3 shows, the multiple linear regression analysis found a low association between attitude scores and having less than a bachelor’s degree (β −0.109; 95% CI −2.541, −0.451) and having experienced the risk of infection but not isolating (β −0.154; 95% CI −4.152, −0.670) than having a bachelor’s degree or above and never having been at risk of contracting COVID-19, respectively.

There was a closer association between satisfaction scores and being married than with being single (β 0.074; 95% CI −0.073, 3.022). In addition, there was a lower association between not being a healthcare worker (β −0.143; 95% CI −4.698, −0.831) and having experienced the risk of infection but not isolating (β −0.132; 95% CI −6.034, −0.502) than being a healthcare worker and never having been at risk of contracting COVID-19.

### 3.3. Correlation between Attitude and Satisfaction Scores

Table 4 indicates a weak but significant positive correlation between attitude and satisfactions scores (*r* = 0.338, *p* < 0.001).

## 4. Discussion

This study was conducted during the third wave of the COVID-19 pandemic in Thailand. To the best of our knowledge, it is the first study to explore COVID-19 vaccine attitudes and satisfaction among participants living in an urban community. We found that socio-demographic factors such as marital status, education, occupation, and risk of contracting COVID-19 influenced participants’ attitudes and satisfaction scores. In addition, we found that attitude scores significantly increased along with satisfaction scores.

All participants reported receiving a first COVID-19 vaccine dose. Our finding that most participants had received a viral vaccine (52.8%) and had never been at risk of contracting COVID-19 differs from previous reports that the likelihood of becoming vaccinated decreases with the increase in adverse events [12]. However, this study is based on the current situation during the COVID-19 pandemic and vaccine distribution in Bangkok, Thailand. The Thai government is among many that have actively promoted vaccine boosters to the population. In doing so, the Thai government has allowed some large companies, universities, and public or semi-public bodies to administer vaccinations.

### 4.1. Vaccine Attitudes: Main Results and the Association with Socio-Demographic Variables

Our study found that most participants agreed with receiving two full doses of the vaccine as prescribed by the state. One study recommended that individuals should receive a second dose of the inactivated vaccine with possible booster doses [13].

Our finding that most participants agreed that obtaining accurate information about the COVID-19 vaccine would influence their decision on the type of vaccine they would receive aligns with the results of a previous study that found that belief that the preparations were safe was a stronger predictor of the willingness to become vaccinated than the fear of becoming ill [14]. Although misinformation about the adverse effects of COVID-19 vaccines has gained traction in segments of social and mass media, the robust VAERS (Vaccine Adverse Events Reporting System) has documented few serious adverse events that were causally linked to vaccination [15]. In addition, our study stronger than previously reported that the most common reasons why the respondents refused to be vaccinated are lack of confidence in the effectiveness of the booster dose and the occurrence of adverse events in them or their loved ones [16].

Our study found that participants agreed that all Thai people and members of the general public living in at-risk areas should receive a third vaccine dose due to the emergence of COVID-19 mutations. A previous study reported on the effectiveness of the viral vector vaccine and the mRNA vaccine and found significant and positive correlations between mixing and matching both vaccines against the Delta variant (*p*-value < 0.010) [17].

In addition, our study found that 40.5% of participants were not sure that government measures would be effective in controlling the spread of COVID-19. Another study reported that countries where vaccine acceptance exceeded 80% tended to be Asian nations with strong trust in the central government (China, South Korea, and Singapore) [18]. Approximately 49% of the participants in this study were not sure that people living in at-risk areas should receive the viral vector vaccine as a third dose; however, another study suggested that the key advantage of viral vector vaccines is that the immunogen is expressed in the context of a heterologous viral infection, which induces the innate immune responses required for the adaptive immune responses [19]. Currently, there is a lack of studies describing adverse events after booster doses [16].

Our multiple linear regression analysis found that a lower association between attitude scores and having less than a bachelor’s degree. A previous study conducted in Poland found little difference in vaccination attitudes based on education [12]; however, an Italian study reported a significant association between higher education and positive beliefs about vaccination [20]. We suggest that future communication efforts targeting those with lower education levels are needed to reduce communication inequalities during pandemic situations.

### 4.2. Vaccine Satisfaction: Main Results and the Association with Socio-Demographic Variables

The majority of participants in this study were satisfied with selected viral vector vaccines in general and with their type and efficacy. One study found that a reason given by participants refusing or delaying the booster dose was their lack of faith in its effectiveness or the belief that the basic vaccination schedule ensures an adequate and long-lasting level of protection [16].

Considerable efforts have been devoted to developing effective and safe vaccines against COVID-19 [19]; however, 41% of this study’s participants expressed only medium satisfaction with the quality of the allocated vaccines and with the type and efficacy of the inactivated vaccine. In Japan, although the government’s vaccine campaign has sought to promote vaccination for all people, around 30% of the Japanese are unsure or neutral regarding vaccine acceptance [21].

In addition, 65% participants in this study expressed low satisfaction that the inactivated vaccine was sufficient for the Thai population and 42% reported low satisfaction that the viral vector vaccine was sufficient. Another study presented that COVID-19 vaccine booster doses include the third dose of two-dose vaccines, such as mRNA vaccines, or the second dose of single-dose vaccines such as viral-vector-based vaccines [4]. Our study found that 53% of participants reported low satisfaction that each vaccine was being distributed across all areas, and 57.3% expressed low satisfaction with the role of government credibility in procuring and administering vaccines. Notably, the Thai government has reported that 75% of the population has received a first vaccination dose, 69.5% has received a second dose, and 18.8% have received a third or booster dose [22].

Our finding of a closer association between vaccine satisfaction among married participants agree with those of previous studies, indicating that married individuals tend to be more likely to accept the vaccine [21,23,24]. One study suggests that this result may be in part due to the emotional and financial burdens associated with divorce [25]. In addition, we found a lower association between non-HCW occupations and COVID-19 vaccine satisfaction, which supports a previous report that HCWs demonstrated significantly greater willingness to undergo vaccination [26].

### 4.3. Correlation between Vaccine Attitudes and Satisfaction

Research has suggested that policymakers work to increase positive emotions in the public to more effectively fight against COVID-19 [27]. Our analysis revealed a weak but significant positive correlation between attitude and satisfaction scores. One study found that lower rates of negative attitudes toward the COVID-19 vaccine than positive attitudes [28]. Another study that examined the reasons for negative attitudes toward vaccination reported that 83% of participants claimed that the vaccine is unsafe, and 84% expressed that there was no need to be vaccinated because they had natural immunity [20]. In addition to the lack of trust, it has been found that vaccine attitudes can be affected by conspiracy theories, lack of knowledge, and anxiety [29].

The Thai government supports vaccination boosters for the entire population [22]. The best of our results agrees with a previous study that determined that the health system informing the public was an important positive determinant of vaccine attitudes and behaviors [30].

### 4.4. Limitations

This study was limited by a number of factors, including that it was conducted online using self-reported data, which could have introduced selection bias. We did not collect information about participants’ job positions, general medical anamnesis and BMI, or side-effects experienced following primer doses. We did not incorporate participants’ perceptions of whether the country could win the fight against the COVID-19 pandemic through implementing social distancing and other restrictive measures. To avoid this issue, the authors distributed the questionnaire not only among groups related to vaccinations and COVID-19 in Thailand but also among healthcare worker groups. This study used a snowball technique, which might have selected bias since people refer those whom they know and have similar traits with. This sampling method can therefore have a potential sampling bias and a margin of error. The population was representative of the wider population of population residing in only one area of Bangkok. Other limitations might be higher proportion of females with a high education, working in healthcare sector, and all vaccinated with at least one dose. Finally, most of participants reporting receiving a first COVID-19 vaccine dose might be related with selection bias.

## 5. Conclusions

This study found a significant positive correlation between vaccine attitudes and satisfaction in a population in urban Bangkok, Thailand. It was determined including that participant (i) agreed that getting accurate information about the COVID-19 vaccine would influence their decision on the type of vaccine they received; (ii) agreed that member of the general public living in at-risk areas should receive a third vaccine dose; and (iii) were satisfied with selecting the viral vector vaccine and with the type and efficacy of the viral vector vaccine. It is important to monitor changes in people’s vaccine acceptability as the vaccine development process continues. Furthermore, arrangements should be made to develop effective strategies to raise awareness about the importance of vaccination. Hence, the Thai government is among many that have actively promoted vaccine boosters to the population. Further observation and evaluation of social attitudes toward vaccination is necessary to inform efforts to ensure effective implementation of the vaccination program in Thailand.

## Figures and Tables

**Table 1 ijerph-19-05086-t001:** Socio-demographic characteristics of participants (*n* = 780).

Category	*n* (%)
**Gender**	
Male	188 (24.1)
Female	592 (75.9)
**Age** *	
≤42 years	395 (50.6)
>42 years	385 (49.4)
**Marital status**	
Single	399 (51.2)
Married	309 (39.6)
Divorced	72 (9.2)
**Education**	
Primary school	37 (4.7)
High school	88 (11.3)
Diploma	90 (11.5)
Bachelor’s	476 (61.0)
Above bachelor’s	89 (11.4)
**Occupational** **status**	
Healthcare worker	387 (49.6)
Government office	140 (17.9)
Private office	70 (9.0)
Non-office employee	32 (4.1)
Personal business	75 (9.6)
Housewife	76 (9.7)
**Disease**	
No diseases	500 (64.1)
Non-communicable disease	280 (35.9)
**Risk of contracting COVID-19**	
Infection risk: isolated	215 (27.6)
Infection risk: not isolated	147 (18.8)
Not sure	79 (10.1)
Never at risk	339 (43.5)
**Vaccination booster history**	
First dose	
None	0
Inactivated vaccine	349 (44.7)
Viral vaccine	412 (52.8)
mRNA vaccine	19 (2.4)
Second dose: booster	
None	21 (2.7)
Inactivated vaccine	349 (44.7)
Viral vaccine	386 (49.5)
mRNA vaccine	24 (3.1)
Third dose: booster	
None	414 (53.1)
Inactivated vaccine	21 (2.7)
Viral vaccine	120 (15.4)
mRNA vaccine	225 (28.8)

* age (mean ± SD) = 42 ± 17.73.

**Table 2 ijerph-19-05086-t002:** Frequency scores for attitude and satisfactions items.

Attitude Items		*n* (%)		
		Disagree	Not Sure	Agree
A1	You have received two doses of the vaccine as prescribed by the state. You think that this will not prevent you from contracting COVID-19.	79 (10.1)	216 (27.7)	485 (62.2)
A2	Although you have received two full doses of the vaccine as prescribed by the state, the best prevention against contracting COVID-19 is to always use a mask, to clean your hands with soap and water or alcohol gel after touching things, and practice social distancing.	31 (4.0)	43 (5.5)	706 (90.5)
A3	You have good knowledge about each type of COVID-19 vaccine, such as inactivated, viral vector, and mRNA vaccines.	41 (5.3)	349 (44.7)	390 (50.0)
A4	Besides vaccinating against COVID-19, recording travel information in and out of different areas through the mobile applications “Mor Chana” and “Thai Chana” are effective for the prevention and surveillance of disease spread?	113 (14.5)	258 (33.1)	409 (52.4)
A5	The vaccination registration system through the application “Doctor Ready” is a simple and hassle-free system.	159 (20.4)	232 (29.7)	389 (49.9)
A6	Getting accurate information about the COVID vaccine will influence your decision on the type of vaccine you will obtain?	47 (6.0)	101 (12.9)	632 (81.0)
A7	You have obtained information about the correct vaccine. Most of the medical and public health professionals are informed or educated about toxic vaccines through online media.	54 (6.9)	149 (19.1)	577 (74.0)
A8	Vaccination is the best protection against COVID-19.	93 (11.9)	233 (29.9)	454 (58.2)
A9	The government’s clear and accurate information and media will be effective in controlling the spread of COVID-19.	123 (15.8)	246 (31.5)	411 (52.7)
A10	You have confidence that government measures will be effective in controlling the spread of COVID-19.	228 (29.2)	316 (40.5)	236 (30.3)
A11	Members of the general public living in at-risk areas should receive a third vaccine dose	42 (5.4)	84 (10.8)	654 (83.8)
A12	People in at-risk areas should be vaccinated with the mRNA vaccine as a third dose?	45 (5.8)	192 (24.6)	543 (69.6)
A13	All Thai people should get a third vaccine dose due to COVID-19 mutations.	44 (5.6)	95 (12.2)	641 (82.2)
A14	People in at-risk areas should be vaccinated should be vaccinated with the viral vector as a third dose	189 (24.2)	381 (48.8)	210 (26.9)
	Attitude mean ± SD = 34.63 ± 6.14			
	“Favorable attitudes” (score ranging 33–42) represent to “Agree” scaled to “3” “Medium favorable attitudes” (score ranging 24–32) represent to “Not sure” scaled to “2” “Unfavorable attitudes” (score ranging 14–23) represent to “Disagree” scaled to “1”			
**Satisfaction Items**		***n* (%)**		
		**Low**	**Medium**	**High**
S1	I am satisfied with the inactivated vaccine.	511 (65.5)	200 (25.6)	69 (8.8)
S2	I am satisfied with the viral vector vaccine.	194 (24.9)	354 (45.4)	232 (29.7)
S3	I have confidence in the quality of the allocated vaccines.	282 (36.2)	318 (40.8)	180 (23.1)
S4	I have a good knowledge and understanding of vaccines that have been identified as being of good quality.	169 (21.7)	398 (51.0)	213 (27.3)
S5	If possible, I will select the viral vector vaccine	189 (24.2)	235 (30.1)	356 (45.6)
S6	The inactivated vaccine is enough for the Thai population	344 (44.1)	296 (37.9)	140 (17.9)
S7	The viral vector vaccine is enough for the Thai population	331 (42.4)	312 (40.0)	137 (17.6)
S8	Registration for injections of the inactivated vaccine will be appointed on time (No delay)	346 (44.4)	309 (39.6)	125 (16.0)
S9	Registration for injections of the viral vector vaccine will be appointed on time (No delay)	271 (34.7)	313 (40.1)	196 (25.1)
S10	Each vaccine is evenly distributed across all areas.	412 (52.8)	277 (35.5)	91 (11.7)
S11	I am satisfied with the role of government credibility in procurement administering vaccines	447 (57.3)	244 (31.3)	89 (11.3)
S12	I am satisfied with the type and efficacy of the inactivated vaccine.	298 (38.2)	371 (47.6)	111 (14.2)
S13	I am satisfied with the type and efficacy of the viral vector vaccine	149 (19.1)	266 (34.1)	365 (46.8)
S14	I am satisfied with the type and efficacy of the mRNA vaccine.	167 (21.4)	323 (41.4)	290 (37.2)
S15	I am satisfied with the cost of the vaccine	330 (42.3)	365 (46.8)	85 (10.9)
	Satisfaction mean ± SD = 26.37 ± 9.69			
	“Low satisfaction” (score ranging 15–25) represent to scaled “1”. “Medium satisfaction” (score ranging 26–35) represent to scaled “2”. “High satisfaction” (score ranging 36–45) represent to scaled “3”.			

**Table 3 ijerph-19-05086-t003:** Association between socio-demographic variables and attitude and satisfaction scores.

Factors Variable	Multiple Linear Regression			
	Attitude		Satisfactions	
	β (95% CI)	*p*-Value	β (95% CI)	*p*-Value
Gender				
Male	Ref.		Ref.	
Female	0.022 (−0.851, 1.480)	0.596	−0.026 (−2.431, 1.273)	0.540
Marital status				
Single	Ref.		Ref.	
Married	0.068 (−0.120, 1.828)	0.086	0.074 (−0.073, 3.022)	0.062
Divorced	−0.063 (−2.999, 0.314)	0.112	−0.029 (−3.611, 1.654)	0.466
Education				
≥Bachelor degree	Ref.		Ref.	
<Bachelor degree	−0.109 (−2.541, −0.451)	0.005	−0.028 (−2.276, 1.046)	0.468
Occupational status				
HCWs	Ref.		Ref.	
Non-HCWs	−0.070 (−2.078, 0.355)	0.165	−0.143 (−4.698, −0.831)	0.005
Risk of contracting COVID-19				
Never at risk	Ref.		Ref.	
Not sure	−0.049 (−2.109, 0.896)	0.429	−0.029 (−2.960, 1.816)	0.639
Risk of infection: not isolated	−0.154 (−4.152, −0.670)	0.007	−0.132 (−6.034, −0.502)	0.021
Risk of infection: isolated	−0.107 (−3.131, 0.196)	0.084	−0.096 (−4.731, 0.556)	0.121
Disease status				
No diseases	Ref.		Ref.	
Non-communicable disease	0.047 (−0.345, 1.538)	0.214	0.024 (−1.019, 1.973)	0.531

Data were analyzed using the multiple linear regression models. Data are presented as β coefficients and 95% confidence intervals (CI).

**Table 4 ijerph-19-05086-t004:** Spearman correlation analysis of attitude and satisfaction scores.

Correlation Coefficient (*r*)			
Variables	Attitudes	Satisfaction	*p*-Value
Attitudes	1.000	0.338	<0.001
Satisfactions	0.338	1.000	<0.001

Correlation coefficient is significant at *p*-value < 0.05.

## Data Availability

The data sets generated and analyzed during the current study are not publicly available due to identifiable information but are available from the corresponding author on reasonable request answering the survey.

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
