# Peer review of "The Relationship between Attitudes and Satisfaction Concerning the COVID-19 Vaccine and Vaccine Boosters in Urban Bangkok, Thailand: A Cross-Sectional Study"

_ijerph, 2022, doi:10.3390/ijerph19095086_

Round 1
Reviewer 1 Report
Dear authors
I appreciate the effort to explore the important issue of vaccine hesitancy.
However, I think that a further effort needs to be made in order to increase the informative value of the project.
Specifically:
Abstract, result section and Introduction: it would be useful to state clearer the distinction between first and second dose to complete the vaccine cycle and booster dose. Similarly, the sentence “Following reports of a gradual decline in protection following vaccination and the positive effects of a booster dose six months after primary vaccination, many countries have made the decision to administer booster doses” may include a short general description of the primary vaccination schedule
Study design and data collection: What do you mean with GPs participants? GPs themselves or their patients? How have you firstly recruited GPs and HCW? It is not clear whether you used some personal contact (i.e. e mails) or social media etc… Please specify.
Questionnaire: How did you obtain a total score ranging from 0 to 42 for the 17 attitude items? How did you identify the three categories, namely low , medium and high satisfaction (i.e. how did you select the boundaries scores)? Did you calculate a Cronbach’s alpha coefficient for the whole questionnaire? Why you did not perform a specific analysis to validate the identified dimensions (e.g. Satisfaction and attitudes)?
Result section. Please justify the following sentence: “These results indicate that the population was representative of the wider population of population residing in Bangkok”. Looking at table 1 results it seems that there is an important selection bias instead.
Table 1 need to be re-written in order to be easier to read (the distinction between questions and possible multiple answers is not clear and this make it difficult to be read)
Table 1 needs to be more informative in order to identify how results can be generalized to the whole population. Specifically, age needs to be described as mean and standard deviation.
An important limitation (as stated in the specific section) is related to selection bias. Your population include a higher proportion of females, with a high education, working in Healthcare Sector and all vaccinated with at least one dose (I suppose that also age represent an important source of bias but I need to have mean and SD). In the discussion section these characteristics need to be more clearly stated in order to give the right interpretation to descriptive results.
Discussion: consider to change the subtitles as follow:
“Associations between socio-demographic variables and vaccine attitudes”: change with: “Vaccine attitudes: main results and the association with socio-demographic variables” (the same for vaccine satisfaction)
As suggested previously, please make clear that your result refer to a specific selected population.
Conclusions are very general. Results should guide vaccine promotion campaign. Can you strongly link results with specific instruction for vaccine promotion considering your selection bias? Otherwise the whole study results quite weak.
Author Response
Reviewer 1
Comments and Suggestions for Authors
Dear authors
Specifically:
Abstract:
Result section and Introduction: it would be useful to state clearer the distinction between first and second dose to complete the vaccine cycle and booster dose. Similarly, the sentence “Following reports of a gradual decline in protection following vaccination and the positive effects of a booster dose six months after primary vaccination, many countries have made the decision to administer booster doses” may include a short general description of the primary vaccination schedule
Authors:
I have incorporated all of your suggestions into my revision. They were very helpful.
The revised paper we have added the following sentences “Hence a gradual decline in protection following vaccination and the positive effects of a booster dose after primary vaccination have made the decision to administer booster doses” in “Abstract”.
Study design and data collection:
What do you mean with GPs participants? GPs themselves or their patients?
Authors:
From study design section “This study entailed the analysis of a cross-sectional survey distributed among GP participants and HCWs in urban Bangkok”: change to: “This study entailed the analysis of a cross-sectional survey distributed among participants in urban Bangkok”.
I would like to explain that GPs are themselves.
The revised paper we have added the following sentences in “study design section”.
How have you firstly recruited GPs and HCW? It is not clear whether you used some personal contact (i.e. e mails) or social media etc… Please specify.
Authors:
I would like to explain that; Firstly, participants aged 18 years and older living in Bangkok. Secondly, after questionnaire data collection by Google form for social media (n=780) were classified into two categories 1) HCWs and 2) Non-HCWs (inducing government office, private office, non-office employee, personal business, and housewife) represent to occupational status.
Questionnaire: How did you obtain a total score ranging from 0 to 42 for the 17 attitude items?
Authors:
The attitude section consists of 14 items scored as Agree = 2, not sure = 1, and disagree = 0. Attitude item responses were summed to a total score ranging from 0–42 and classified into three categories, namely favorable attitudes (33-42), medium favorable attitudes (24-32), and unfavorable attitudes (14-23) (see Table 2 for details).
Explains that;
- “Favorable attitudes” (score ranging 33-42) represent to “Agree” scaled to “3”
- “Medium favorable attitudes” (score ranging 24-32) represent to “Not sure” scaled to “2”
- “Unfavorable attitudes” (score ranging 14-23) represent to “Disagree” scaled to “1”
Thus, attitudes scaled 0, 1, 3 were classified into three categories for represent attitude item responses.
The revised paper we have added the following sentences in “Questionnaire” and Table 2.
How did you identify the three categories, namely low, medium and high satisfaction (i.e. how did you select the boundaries scores)?
Authors:
The satisfaction section consists of 15 items scored as very satisfied = 3, medium satisfied = 2, and low satisfied = 1. Satisfaction item responses were summed to a total score ranging from 0–45 and classified into three categories, namely low satisfaction (15–25), medium satisfaction (26–35), and high satisfaction (36–45) (see Table 2 for details).
Explains that;
- “Low satisfaction” (score ranging 15–25) represent to scaled “1”.
- “Medium satisfaction” (score ranging 26–35) represent to scaled “2”.
- “High satisfaction” (score ranging 36–45) represent to scaled “3”.
Thus, satisfaction scaled 0, 1, 3 were classified into three categories for represent satisfaction item responses
The revised paper we have added the following sentences in “Questionnaire” and Table 2.
Did you calculate a Cronbach’s alpha coefficient for the whole questionnaire? Why you did not perform a specific analysis to validate the identified dimensions (e.g. Satisfaction and attitudes)?
Authors:
I would like to explains that: Content validity and Cronbach’s alpha coefficient of satisfaction and attitudes were calculated separately. In addition, was examined by three experts.
Cronbach’s alpha coefficient for attitudes showed 0.91 and content validity 0.87.
Cronbach’s alpha coefficient for satisfaction showed the item objective congruence (IOC) score was 0.93. This study reported a Cronbach’s alpha coefficient of 0.751.
The revised paper we have added the following sentences in “Questionnaire section” that “Content validity was examined by three experts, Cronbach’s alpha coefficient for attitudes showed 0.91 and content validity 0.87. For the satisfaction showed the item content validity score was 0.93, a Cronbach’s alpha coefficient of 0.751”.
Result section.:
Please justify the following sentence: “These results indicate that the population was representative of the wider population of population residing in Bangkok”. Looking at table 1 results it seems that there is an important selection bias instead.
Authors:
I would like to explain that this study we focus urban community Bangkok only, which is policy for Faculty of medicine.
However, I have incorporated all of your suggestions into my revision. They were very helpful.
Thus, we added “The population was representative of the wider population of population residing in Bangkok” from you mention in “limitation section”.
The revised paper we have added the following sentences in “Limitations section”.
Table 1 need to be re-written in order to be easier to read (the distinction between questions and possible multiple answers is not clear and this make it difficult to be read). Table 1 needs to be more informative in order to identify how results can be generalized to the whole population.
Authors:
I have incorporated all of your suggestions into my revision. They were very helpful.
First of all, we would like to explains that participants aged 18 years and older living in Bangkok. Second, after questionnaire data collection by Google form for social media (n=780) were classified into two categories 1) HCWs and 2) Non-HCWs (inducing government office, private office, non-office employee, personal business, and housewife) represent to occupational status.
In study design section we would like to change that “This study entailed the analysis of a cross-sectional survey distributed among GP participants and HCWs in urban Bangkok”: change with: change with: “This study entailed the analysis of a cross-sectional survey distributed among participants in urban Bangkok”.
The revised paper we have added the following sentences in “study design section”.
#
Thus, this study we classified population into two groups after data collection including
For table 1., we would like to presented what kind of participants represent by “Occupational status”
In addition, for multiple linear regression were classified into two categories 1 = HCWs and 2 = Non-HCWs (inducing government office, private office, non-office employee, personal business, and housewife) represent to occupational status.
Specifically, age needs to be described as mean and standard deviation.
Authors:
age (mean ± SD) = 42±17.73
This is the reason why we classified into two categories that ≤ 42 years and > 42 years.
The revised paper we have added the following sentences age (mean ± SD) = 42±17.73 in “bottom table 1”
An important limitation (as stated in the specific section) is related to selection bias. Your population include a higher proportion of females, with a high education, working in Healthcare Sector and all vaccinated with at least one dose (I suppose that also age represent an important source of bias but I need to have mean and SD).
Authors:
I have incorporated all of your suggestions into my revision. They were very helpful.
The revised paper we have added the following sentences that “Other limitations might be higher proportion of females, with a high education, working in healthcare sector and all vaccinated with at least one dose” in “Limitations section”.
The revised paper we have added the following sentences age (mean ± SD) = 42±17.73 in “bottom table 1”.
In addition, this study based on real situation COVID-19 pandemic. However, individuals who have already been vaccinated may accept or reject a vaccine booster dose due to various reasons, including side effects experienced following the previous (primer) doses, the perceived effectiveness of the vaccine booster dose, the vaccinee’s susceptibility to the target infection, and other safety concerns.
From this evident lend to objective that to assess attitudes and satisfaction concerning COVID-19 vaccines and vaccine boosters in the population in Bangkok, Thailand.
Moreover, all vaccinated with at least one dose because “The Thai government is among many that have actively promoted vaccine boosters to the population” we added in “Discussion section”
In the discussion section these characteristics need to be more clearly stated in order to give the right interpretation to descriptive results. Discussion: consider to change the subtitles as follow: “Associations between socio-demographic variables and vaccine attitudes”: change with: “Vaccine attitudes: main results and the association with socio-demographic variables” (the same for vaccine satisfaction)
Authors:
I have incorporated all of your suggestions into my revision.
The revised paper we have added the following sentences in “Discussion section” that:
- “Vaccine attitudes: main results and the association with socio-demographic variables”
- “Vaccine satisfaction: main results and the association with socio-demographic variables”
As suggested previously, please make clear that your result refer to a specific selected population.
Authors:
From study design section in “This study entailed the analysis of a cross-sectional survey distributed among GP participants and HCWs in urban Bangkok”: change with: “This study entailed the analysis of a cross-sectional survey distributed among participants in urban Bangkok”.
The revised paper we have added the following sentences in “study design section”.
Conclusions are very general. Results should guide vaccine promotion campaign. Can you strongly link results with specific instruction for vaccine promotion considering your selection bias? Otherwise the whole study results quite weak.
Authors:
From you mention that “Results should guide vaccine promotion campaign”: we would like to explain that “The Thai government is among many that have actively promoted vaccine boosters to the population” we added in “Discussion section” and hence in “conclusions section”
From you mention that “Can you strongly link results with specific instruction for vaccine promotion considering your selection bias” we added in “Limitation section” that “Finally, most of participants reported receive a first COVID-19 vaccine dose might be related with selection bias”.
From you mention that “Otherwise the whole study results quite weak” we added in … Vaccine attitudes: main results and the association with socio-demographic variables section that “In addition, our study stronger than previously reported that the most common reasons why the respondents refused to be vaccinated are lack of confidence in the effectiveness of the booster dose and the occurrence of adverse events in them or their loved ones [Ref. 19].”
Moreover, in “correlation between vaccine attitudes and satisfaction” section we conclude that “The Thai-government supports vaccination boosters for the entire population [Ref. 22].”

Reviewer 2 Report
The paper presents questionnaire research concerning attitudes and satisfactions about Covid vaccines. The research may be interesting, and generally, it may be accepted for publication.
I have only some minor remarks and objections:
- Please explain why Attitudes are scaled 0,1,2, and Satisfaction is scaled to 1,2,3
The results do not support some conclusions, e.g., Page 6: "closer association between satisfaction scores and being married".
Here, we have p-value=0.06, Is it statistically significant? A small correction for multiple tests should be used here. The authors build two models, each with seven variables, resulting in 14 tests.
- Why only a subset of variables presented in Tab.1 are used in linear models.
- I suggest adding histograms for outputs: Attitude and Satisfaction.
Author Response
Reviewer 2
Comments and Suggestions for Authors
The paper presents questionnaire research concerning attitudes and satisfactions about Covid vaccines. The research may be interesting, and generally, it may be accepted for publication.
I have only some minor remarks and objections:
Authors:
I have incorporated all of your suggestions into my revision. They were very helpful.
- Please explain why Attitudes are scaled 0,1,2, and Satisfaction is scaled to 1,2,3
Authors:
The attitude section consists of 14 items scored as Agree = 2, not sure = 1, and disagree = 0. Attitude item responses were summed to a total score ranging from 0–42 and classified into three categories, namely favorable attitudes (33-42), medium favorable attitudes (24-32), and unfavorable attitudes (14-23) (see Table 2 for details).
Explains that;
- “Favorable attitudes” (score ranging 33-42) represent to “Agree” scaled to “3”
- “Medium favorable attitudes” (score ranging 24-32) represent to “Not sure” scaled to “2”
- “Unfavorable attitudes” (score ranging 14-23) represent to “Disagree” scaled to “1”
Thus, attitudes scaled 0, 1, 3 were classified into three categories for represent attitude item responses
The satisfaction section consists of 15 items scored as very satisfied = 3, medium satisfied = 2, and low satisfied = 1. Satisfaction item responses were summed to a total score ranging from 0–45 and classified into three categories, namely low satisfaction (15–25), medium satisfaction (26–35), and high satisfaction (36–45) (see Table 2 for details).
Explains that;
- “Low satisfaction” (score ranging 15–25) represent to scaled “1”.
- “Medium satisfaction” (score ranging 26–35) represent to scaled “2”.
- “High satisfaction” (score ranging 36–45) represent to scaled “3”.
Thus, satisfaction scaled 0, 1, 3 were classified into three categories for represent satisfaction item responses
The revised paper we have added the following sentences in “Questionnaire” and “Table 2”.
- The results do not support some conclusions, e.g., Page 6: "closer association between satisfaction scores and being married". Here, we have p-value=0.06, Is it statistically significant?
Authors:
I would like to explain that p-value as 0.06 in statistical term is not significantly, but in other hand p-value nearly 0.05 we can use word inducing “nearly or closer”
Thus, our study presented that “There was a closer association between satisfaction scores and being married than with being single (ᵦ 0.074; 95% CI -0.073, 3.022)” or another explain that not significantly but nearly or closer statistically significant.
- A small correction for multiple tests should be used here. The authors build two models, each with seven variables, resulting in 14 tests.
Why only a subset of variables presented in Tab.1 are used in linear models.
Authors:
I would like to explain that a bivariate analysis of each variable was first done, and then variables with p- value < 0.25 were included in the multivariate analysis.
Thus, a bivariate analysis results found that Gender, Marital status, Education, Occupational status, Risk of contracting COVID-19 and Disease status were showed p- value < 0.25.
Hence, they variables were used in Multiple linear regression model.
I suggest adding histograms for outputs: Attitude and Satisfaction.
Authors:
Histogram for attitude. We concluded that attitude mean ± SD = 34.63 ± 6.14.
Thus, these results were presented in Table 2.
………………………………………………………………………………………………………………………………………………..
Histogram for satisfaction. We concluded that satisfaction mean ± SD = 26.37 ± 9.69.
Thus, these results were presented in Table 2.
Hence, from you mention that “should be added histograms for outputs: Attitude and Satisfaction”
In the revised paper we have added the following sentence in Table 2.
We hope you sympathize and agree with this.

Round 2
Reviewer 1 Report
Dear Authors thank you for your efforts in improving the manuscript.
I have no others suggestion.